# An Efficient 4H-SiC Photodiode for UV Sensing Applications

**Mohamed L. Megherbi [1], Hichem Bencherif [2]** (ID)**, Lakhdar Dehimi [1], Elisa D. Mallemace [3], Sandro Rao [3,*]** (ID)**,
Fortunato Pezzimenti [3] and Francesco G. Della Corte [4]** (ID)

1   LMSM—Laboratory of Metallic and Semiconducting Materials, University of Biskra, Biskra 07000, Algeria;
    mlmegherbi@yahoo.fr (M.L.M.); la_dehimi@yahoo.fr (L.D.)
2   LAAAS—Laboratory, Mostefa Benboulaid University, Batna 05000, Algeria; hichembencherifeln@gmail.com
3   Department DIIES, Mediterranea University, 89124 Reggio Calabria, Italy; elisa.mallemace@unirc.it (E.D.M.);
    fortunato.pezzimenti@unirc.it (F.P.)
4   Department DIETI, University of Naples Federico II, 80125 Naples, Italy;
    francescogiuseppe.dellacorte@unina.it
*   Correspondence: sandro.rao@unirc.it

**Abstract:** In this paper, we report experimental findings on a 4H-SiC-based p-i-n photodiode. The fabricated device has a p-type region formed by ion-implantation of aluminum (Al) in a nitrogen doped n-type layer. The dark reverse current density reaches 38.6 nA/cm$^2$ at $-10$ V, while the photocurrent density rises to 6.36 μA/cm$^2$ at the same bias under λ = 315 nm ultraviolet (UV) radiation with an incident optical power density of 29.83 μW/cm$^2$. At the wavelength of λ = 285 nm, the responsivity is maximum, 0.168 A/W at 0 V, and 0.204 A/W at $-30$ V, leading to an external quantum efficiency of 72.7 and 88.3%, respectively. Moreover, the long-term stability of the photodiode performances has been examined after exposing the device under test to several cycles of thermal stress, from 150 up to 350 °C and vice versa. The achieved results prove that the examined high-efficiency UV photodiode also has a stable responsivity if subjected to high temperature variations. The proposed device is fully compatible with the conventional production process of 4H-SiC components.

**Keywords:** 4H-SiC; p-i-n photodiode; temperature effect; responsivity

## 1. Introduction

Ultraviolet detectors have significant uses in different areas, from astronomy to combustion control in the automotive industries, etc. [1]. The cost and the volume of the commonly used UV photomultipliers are still high, and the sensitivity to thermal stress remains a critical issue [2]. For these reasons, compact and affordable semiconductors photodiodes are suitable alternatives [3].

Supported by its distinguished properties, such as high thermal conductivity, wide band gap and critical breakdown strength, Silicon Carbide (SiC) is an intriguing choice for UV detection applications, also when the exposure to high temperatures could be an issue [4–7]. SiC photodiodes respond to light from short to long UV waves (200–400 nm), with reduced noise due to visible or infrared radiation [8,9].

Although photodiodes made in 6H-SiC have been examined in earlier time studies [10], the 4H-SiC polytype has gained much more attention in different application fields in the meantime [11,12] thanks to its improved electronic properties [13,14]. 4H-SiC-based photodiodes, in particular Schottky [15,16] and avalanche devices [17–19], have been extensively demonstrated in the last few years; only a few examples are reported instead for p-i-n structures [20–22], although they are more efficient from the optical-electrical conversion point of view.

Hereafter, we report the characteristics of an experimental 4H-SiC p-i-n UV photodiode showing a higher responsivity peak with respect to similar devices that have appeared to date in the literature [15–22]. Moreover, the fabricated devices have been stressed by temperature in order to investigate the effects on the electro-optical characteristics after

a long exposure, in a thermostatic chamber, to cycling temperature ramps, up to 350 °C. Results show that, within this temperature range, the photodiode does not modify its output characteristics also after many days from the first characterization.

## 2. Experiments

The studied and experimentally characterized photodiode is an integral part of a microchip processed by the CNR-Institute for Microelectronics and Microsystem of Bologna (Italy) [23]. Figure 1 shows the device cross-section, together with the main geometric dimensions, of the silicon carbide p-i-n structure considered in this work.

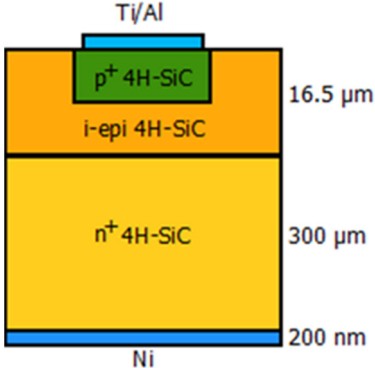

**Figure 1.** Schematic cross-section of a 4H-SiC photodiode.

The p-i-n diodes were manufactured on commercial, 300-micrometer-thick, n+ type 4H-SiC wafers [24], 100 mm in diameter, on which a homo-epitaxial layer with orientation <0001> and conductivity equal to 0.021 $\Omega \cdot$cm was grown. The region that forms the p-type anode, with a circular area of $9.62 \times 10^{-4}$ cm$^2$ (350 μm in diameter), was obtained by ionic implantation of aluminum (Al), while photolithographic processes and chemical etchings were used to create the upper concentric Ti/Al anode electrical contact, $2.41 \times 10^{-4}$ cm$^2$ (175 μm in diameter). Many p-i-n diodes were fabricated in the same microchip with the common cathode consisting of the n+ 4H-SiC substrate. More details about the manufacturing process are given in references [25,26]. To allow easy and stable electrical connections from the chip to the measurement set-up, each anode electrode was contacted by ultrasonic wire-bonding, using a thin Al-wire 50 μm in diameter. Finally, the chip was encapsulated in a custom package for optical and thermal characterization.

## 3. Results and Discussions

The characterization of the 4H-SiC photodiode was performed in a dark box at room temperature (RT) in order to obtain I–V characteristics in the absence of environmental interferences. Figure 2 reports the typical J–V characteristic of the investigated photodiode in forward bias, from 0 to 3 V. From this figure, it is evident that the current shows the set-up of an efficient carrier injection regime starting from a ~1.5 V anode bias.

Moreover, the series resistance, *Rs*, and the ideality factor, $\eta$, have been extracted from the forward J–V characteristics. By fitting the analytic diode equation, the values of *Rs* and $\eta$ are extracted to be 489 $\Omega$ and 1.8, respectively, suggesting a good quality of p-i-n junction [21,27].

Figure 3 shows the J–V characteristic in reverse bias condition from 0 to −30 V. From this figure, an increasing reverse current density J$_D$ is observed with voltage, reaching 22.3 and 78.6 nA/cm$^2$, respectively at −5 and −30 V.

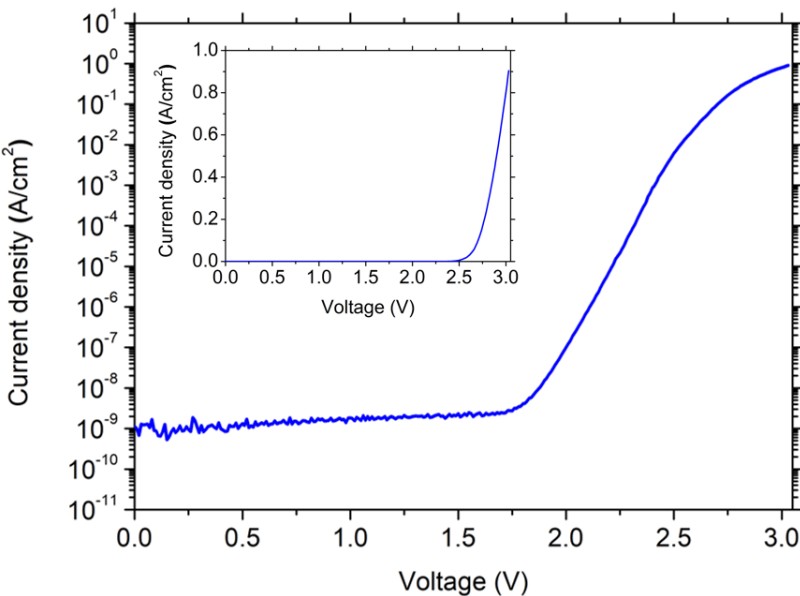

**Figure 2.** Current density vs. voltage characteristic at RT in forward bias from 0 to 3 V, in dark conditions. The insert reports the J–V characteristic in linear scale.

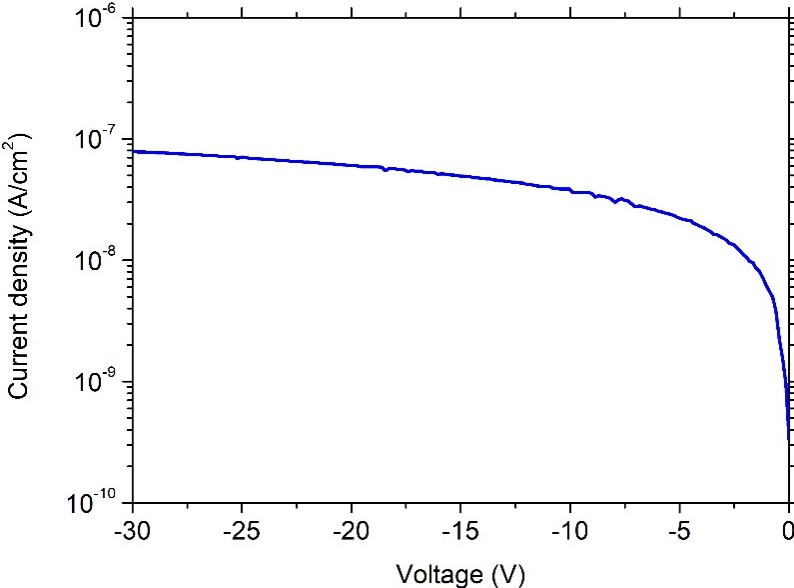

**Figure 3.** Current density–voltage characteristic, in dark conditions at RT, for reverse biases from 0 to −30 V.

By exposing the photodiode to a UV radiation produced by a remotely controlled monochromator, we measured the J–V characteristics at varying wavelengths, as shown in Figure 4 for specific wavelengths and incident optical powers. Electro-optical measurements were performed in the wavelength range between 210 and 380 nm, in steps of 5 nm. As the wavelength increases, we observe that the photocurrent increases, reaching its maximum value at 315 nm. At this wavelength, for which the optical power density at the surface is 29.83 nW/cm$^2$, the total current density is 5.44 µA/cm$^2$ at 0 V, increasing to 6.88 µA/cm$^2$ at −30 V. It should be noted that the photo-generated current amplitude is relevant with respect to the dark current, in spite of the low incident optical power at the surface.

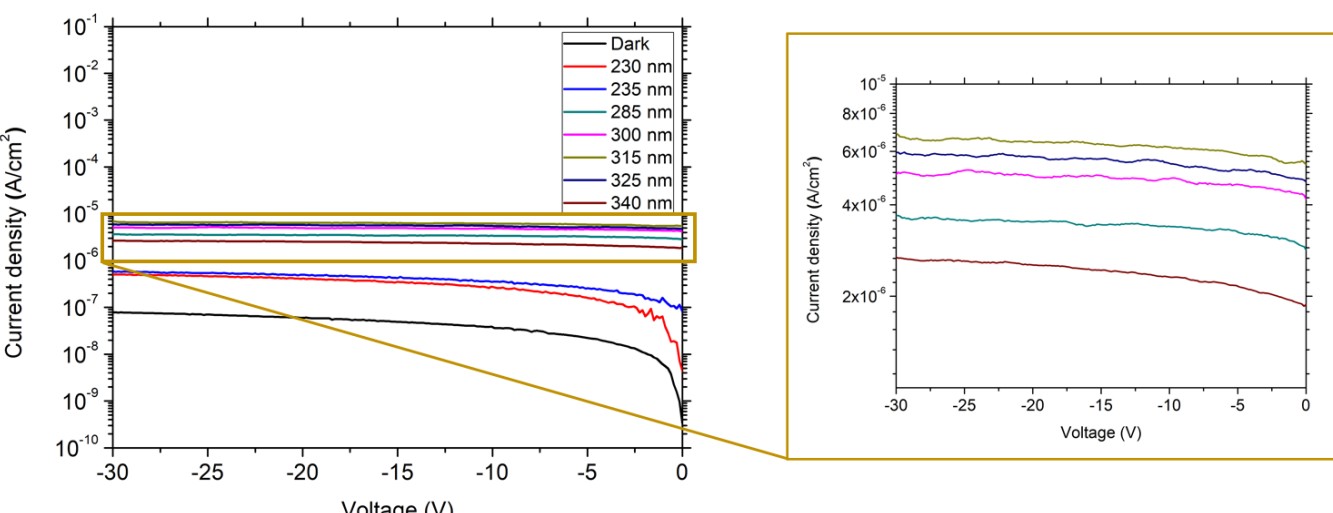

**Figure 4.** J–V characteristics for different UV wavelengths at RT. The incident optical power, from a monochromator, varies with the wavelength.

Figure 5 plots the photocurrent density as a function of the wavelength at 0 and 30 V reverse-biased values.

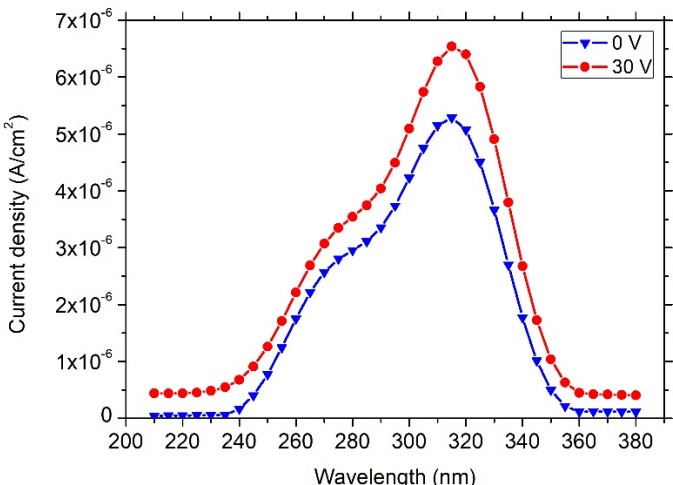

**Figure 5.** Photocurrent density as a function of the wavelength values at 0 and 30 V reverse bias.

The photocurrent shows a peak at a 315-nanometer wavelength; however, this depends on the wavelength-dependent incident optical power. For this reason, a full characterization of the monochromator was performed in order to calculate the optical power spectrum at a distance, from the UV radiation generator output surface, of 7 cm. A commercial DET 210 photodiode [28], with known active area and fully characterized responsivity, R($\lambda$) at all wavelengths, was used. The measured output voltage signal (V) is proportional to the direct photocurrent at the photodiode anode, function of the incident light power (P) and wavelength according to the following:

$$P = \frac{V}{R(\lambda)R_{load}} \tag{1}$$

where $R_{Load}$ is an external load resistance of 50 ohm.

From the calculation of the active area of the 4H-SiC photodiode shown in Figure 1, a circular crown of area $7.21 \times 10^{-4}$ cm$^2$, and the corresponding incident optical power at each wavelength, the responsivity of the 4H-SiC photodiode under the UV radiation from the monochromator has been extracted. The responsivity, that is a figure of merit for

detection, measures the ratio of the photogenerated current ($I_{ph}$) to the incident optical power (P), and can be calculated from the following relation:

$$R = \frac{I_{ph}}{P} \tag{2}$$

The quantum efficiency (η), that measures the number of charge carriers collected per incident photon, has been calculated from responsivity by following formula:

$$\eta = R\frac{h\nu}{e} \tag{3}$$

where hν is the photon energy and e the elementary charge.

Figure 6 shows the spectral responsivity curve, at 0 V bias, in the UV range, from 210 to 385 nm. In the same plot, the corresponding quantum efficiency characteristics are reported.

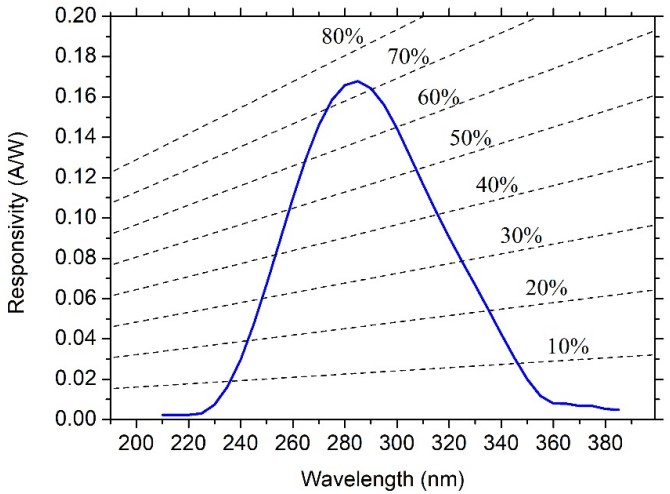

**Figure 6.** Responsivity and quantum efficiency (%) of the 4H-SiC photodiode as a function of wavelength.

To our knowledge, the calculated responsivity peak of 0.168 A/W, found at the wavelength of 285 nm, is the best value ever reported for UV photodiodes with no bias applied. A comparison between Schottky and p-i-n based UV photodiodes, reported in the literature to date, is shown in Table 1.

**Table 1.** Responsivity and Q.E. comparison for Schottky and p-i-n based UV photodiodes reported in the literature to date.

| Ref. | Device | Responsivity Peak (A/W) | Reverse Voltage (V) | Wavelength at Peak (nm) | Q.E. (%) | Thermal Stability |
|---|---|---|---|---|---|---|
| [15] | Schottky | 0.093 | −15 | 270 | 42.5 | n.a. |
| [16] | Schottky | 0.115 | 0 | 285 | 50.0 | 200 °C |
| [20] | p-i-n | 0.096 | 0 | 270 | 44.4 | 175 °C |
| [21] | p-i-n | 0.13 | 0 | 266 | n.a. | 450 K |
| [22] | p-i-n | 0.13 | −5 | 270 | 61 | 180 °C |
| [This work] | p-i-n | 0.168 0.204 | 0 −30 | 285 | 72.7 88.3 | 350 °C |

Figure 7 shows the responsivity peak and quantum efficiency as functions of the reverse bias voltage at λ = 285 nm. It is shown that higher responsivity and quantum efficiency are achieved as the reverse bias increases. This fact is attributed to the expansion of the depletion region that enhances the collection efficiency and, therefore, the separation

mechanism of the photogenerated carriers [29], which, in turn, boosts the device's current capability.

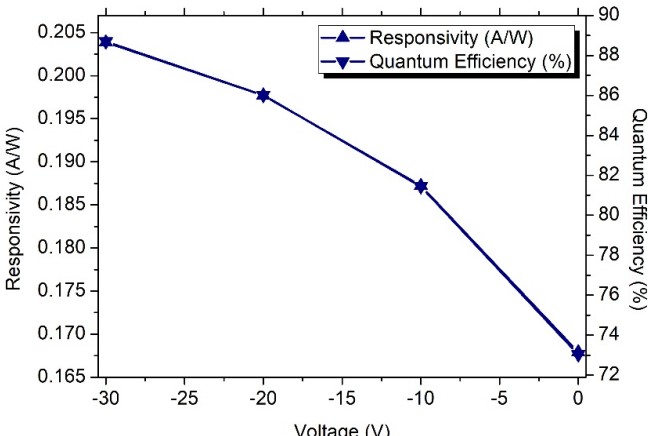

**Figure 7.** Responsivity peak and corresponding quantum efficiency as functions of the reverse bias voltage.

The peak values of both responsivity and quantum efficiency for different biasing voltages are summarized in Table 2.

**Table 2.** Responsivity and quantum efficiency for different bias condition at a 285-nanometer wavelength.

| Reverse Bias | Responsivity Peak (A/W) | Quantum Efficiency (%) |
|---|---|---|
| 0 V | 0.168 | 72.7 |
| 10 V | 0.187 | 81.1 |
| 20 V | 0.198 | 85.6 |
| 30 V | 0.204 | 88.3 |

Finally, the 4H-SiC photodiode has been characterized in terms of viewing angles in a dark room at RT. The photodiode, placed always at the same distance of 7 cm from the monochromator output, was subjected to an UV radiation at the wavelength of λ = 285 nm. The following measurements, shown in Figure 8, were performed with viewing angles between −45° and +45°, in steps of 5°.

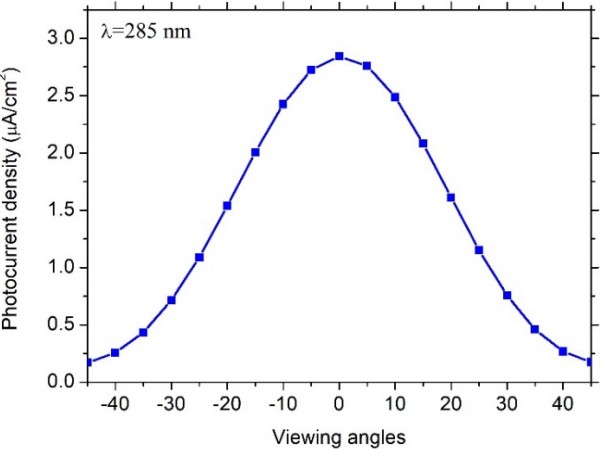

**Figure 8.** 4H-SiC photodiode photocurrent density at varying viewing angles.

## 4. Analysis of High-Temperature Effects

A full characterization of the 4H-SiC photodiode was performed taking into account the effects on the device performances after deep cycles of thermal stress. These measurements were carried out in an adiabatic chamber, where temperatures up to 350 °C are reached.

In particular, ten cycles of measurements were iterated, in a long period of time, from RT up to 350 °C and vice versa, in order to verify the photodiode optical and electrical stability at the end of such thermal stress. Moreover, the same device under test (DUT) has been kept at 350 °C for 24 h, and afterwards, the IV characteristics in dark condition and under UV illumination were measured at RT.

Figure 9 shows the responsivity curves as measured pre- and post the provided thermal solicitation, leading to a very good reproducibility of the spectral photodiode output response.

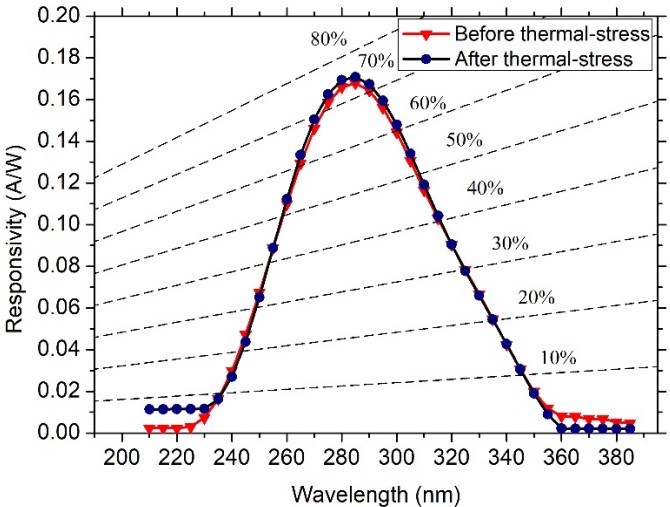

**Figure 9.** 4H-SiC p-i-n photodiode responsivities before and after ten cycles of measurements, from RT up to 350 °C and vice versa. During the last cycle, the DUT has been kept at T = 350 °C for 24 h.

## 5. Conclusions

In this paper, we focused on an investigation of the electro-optical performances related to a 4H-SiC p-i-n photodiode fabricated via an ion implantation technique. At −10 V, a low value of the dark reverse current density is detected, $J_D$ = 38.6 nA/cm$^2$, allowing the photocurrent to be correctly measured under UV radiation. At the wavelength of λ = 285 nm, the photo-response peak is 0.204 A/W at −30 V, providing a better value if compared to those found in the literature, including Schottky photodiode, p-i-n, and more sophisticated bipolar devices. Moreover, the calculated quantum efficiency is 72.7%, the best value ever reported for UV photodiodes with no bias applied.

Furthermore, the photodiode performance has been analyzed after several cycles of thermal stress with temperatures up to 350 °C showing a stable optical response, even after different long time thermal solicitations. This last feature is welcomed by research community and industry, above all when 4H-SiC-based UV detectors must operate in critical applications where the sensitivity to thermal stress is a critical issue.

**Author Contributions:** Conceptualization, S.R. and F.G.D.C.; Data curation, S.R., E.D.M. and F.G.D.C.; Formal analysis, S.R., M.L.M., H.B., L.D. and F.G.D.C.; Investigation, S.R.; Methodology, S.R.; Supervision, S.R. and F.G.D.C.; Validation, S.R. and F.G.D.C.; Writing—original draft, S.R.; Writing—review & editing, S.R., L.D., E.D.M., F.P. and F.G.D.C. All authors have read and agreed to the published version of the manuscript.

**Funding:** This research received no external funding.

**Data Availability Statement:** Data is contained within the article.

**Acknowledgments:** Roberta Nipoti and the Clean Room staff of the CNR-IMM Unit of Bologna (Italy) are thankfully recognized for providing the p-i-n diode and for stimulating discussions.

**Conflicts of Interest:** The authors declare no conflict of interest.

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
