# Peer review of "An Efficient 4H-SiC Photodiode for UV Sensing Applications"

_electronics, doi:10.3390/electronics10202517_

Round 1

Reviewer 1 Report

In the paper An efficient 4H-SiC UV Photodiode for UV sensing applications, authors presented interesting results of SiC UV photodiode. Presented work is interesting and obtained photodetectors have relatively good responsivity. However, in the current version of the paper some issues have to be addressed before publication:

  • lines 35-36: SiC photodiodes respond to light from close- to middle-UV, with reduced noise due to visible or infrared radiation [7,8].  - please specify close and middle UV
  • lines 58-59 please add information about the size of the substrate and its producer
  • lines 72-73: authors used I-V characteristic and J-V characteristic, please decide for one of these
  • Figure 2 and 3 - why authors divided J-V results into two separate characteristics? Usually dark J-V of photosensitive devises is shown in forward and reverse direction on the same graph. See for example photodetector DOI:10.1038/s41699-017-0008-4, figure 2 and solar cell https://doi.org/10.3390/en14154651, figure 7
  • In the case of pn junction based device the knowledge of ideality factor n and series resistance Rs determined from dark J-V characteristic (Figure 2) provides some valuable information about physical phenomena occurring in the junction. In my opinion authors should provide detailed analysis of ideality factor and series resistance o their pristine photodetector and compare the results with already reported.
  • The rectify properties of the junction are often described by RR - rectification ratio, see https://doi.org/10.1016/j.physe.2013.09.001
  • Figure 4 - author wrote: Electro-optical measurements were performed in the wavelength range between 210 nm and 380 nm, in steps of  5 nm. While the step in characteristics plotted in Figure 4 is significantly bigger. Please add J-V characteristics close to optimal wavelength of 315 nm (... 300, 305, 310, 320, 325 ...)
  • line 103 - please use cm or m instead of mm
  • Figures 5 and 4 - the notation of current on Y axis is not the same
  • Figure 4 vs 6: maximum photocurrent was obtain under 315 nm excitation (Figure 4) while maximum responsivity for 285 nm (Figure 6). Please explain that
  • Did authors measured or calculated the quantum efficiency? There is no information about that in the text
  • Table 1 - I would add word Reverse in first column description and please round the numbers of responsivity 
  • Did authors measure some J-V characteristics under UV illumination of the device at the temperature of 350°C?
  • Please add J-V dark characteristics at 350°C and compare n, Rs with RT findings. Alternatively please add J-V characteristics and saturation current, n, Rs analysis after thermal stress test

Reviewer 2 Report

The authors presented a study on the use of 4H-SiC with Al as the p-type layer and N-doped 4H-SiC as the n-type layer for UV sensing application. The use of 4H-SiC as UV sensing is actually not new and there is a lot of published research on this topic. Thus, the novelty and the actual purpose of the paper are a problem. The results are all in conjunction with the previously published papers and there is no new information that can be derived from them. Thus, this paper is not acceptable for publication in the journal.

Round 2

Reviewer 1 Report

I appreciate authors correction of the paper.

Reviewer 2 Report

The authors conducted a study on the use of 4H-SiC with Al as the p-type layer and N-doped 4H-SiC as the n-type layer for UV sensing application. The authors presented 3 components of novelty in terms of structure and compatibility, efficiency, and thermal stability. However, the proposed p-i-n structure is already been presented in published journals. With regards to efficiency, they only presented that 285 nm is better compared to other literature but failed to mention other wavelengths. And lastly, articles having this configuration have reached more than 400 deg C which is still stable. Thus, this publication is not acceptable to be published in this journal.
